# The Prognostic Value of Anemia in Patients with Preserved, Mildly Reduced and Recovered Ejection Fraction

**DOI:** 10.3390/diagnostics12020517

**Published:** 2022-02-17

**Authors:** Anita Pintér, Anett Behon, Boglárka Veres, Eperke Dóra Merkel, Walter Richard Schwertner, Luca Katalin Kuthi, Richard Masszi, Bálint Károly Lakatos, Attila Kovács, Dávid Becker, Béla Merkely, Annamária Kosztin

**Affiliations:** Heart and Vascular Center, Semmelweis University, 1122 Budapest, Hungary; pintera98@gmail.com (A.P.); behona@gmail.com (A.B.); veres.boglarka@semmelweis-univ.hu (B.V.); merkel.eperke@gmail.com (E.D.M.); schwertner.walter_richard@semmelweis-univ.hu (W.R.S.); lucakuthi.cvc@gmail.com (L.K.K.); masszi.richard@gmail.com (R.M.); lakatosbalintka@gmail.com (B.K.L.); kovatti@gmail.com (A.K.); becdavid@gmail.com (D.B.); kosztin.annamaria@gmail.com (A.K.)

**Keywords:** HFpEF, HFmrEF, HFrecEF, HFimpEF, anemia, mortality predictors in heart failure

## Abstract

Data on the relevance of anemia in heart failure (HF) patients with an ejection fraction (EF) > 40% by subgroup—preserved (HFpEF), mildly reduced (HFmrEF) and the newly defined recovered EF (HFrecEF)—are scarce. Patients with HF symptoms, elevated NT-proBNP, EF ≥ 40% and structural abnormalities were registered in the HFpEF-HFmrEF database. We described the outcome of our HFpEF-HFmrEF cohort by the presence of anemia. Additionally, HFrecEF patients were also selected from HFrEF patients who underwent resynchronization and, as responders, reached 40% EF. Using propensity score matching (PSM), 75 pairs from the HFpEF-HFmrEF and HFrecEF groups were matched by their clinical features. After PMS, we compared the survival of the HFpEF-HFmrEF and HFrecEF groups. Log-rank, uni-and multivariate regression analyses were performed. From 375 HFpEF-HFmrEF patients, 42 (11%) died during the median follow-up time of 1.4 years. Anemia (HR 2.77; 95%CI 1.47–5.23; *p* < 0.01) was one of the strongest mortality predictors, which was also confirmed by the multivariate analysis (aHR 2.33; 95%CI 1.21–4.52; *p* = 0.01). Through PSM, the outcomes for HFpEF-HFmrEF and HFrecEF patients with anemia were poor, exhibiting no significant difference. In HFpEF-HFmrEF, anemia was an independent mortality predictor. Its presence multiplied the mortality risk in those with EF ≥ 40%, regardless of HF etiology.

## 1. Introduction

Heart failure (HF) affects 1–2% of the adult population in developed countries, and the prevalence of HF is reaching ≥10% among the elderly [1]. In patients with HF, 22–73% of subjects present with a preserved ejection fraction (HFpEF), which has a wide range of prevalence based on the varying definitions of the disease [1]. Such patients often suffer from multiple concomitant diseases, which also adversely affect their outcomes [2,3,4,5]. Anemia is one of the most frequent comorbidities in this patient population; however, its prevalence in HFpEF patients varies widely, occurring in 21–68% of hospitalized patients, 19–27% in randomized controlled trial participants, and 30–33% in outpatient cohorts [6].

Previous findings suggest a complex, bidirectional relationship between anemia and HF. Between the pathophysiological factors, e.g., inflammation, hemodilution, bone marrow deficiency, nutritional or metabolic factors, and renal dysfunction, each can take a role [7,8]. Anemia may lead to a decrease in the amount of oxygen transport to each cell of the human body, and thus induces a deficit of oxygen in tissues, and anaerobic metabolism, which can have negative consequences, mainly mediated through the activation of the sympathetic and renin-angiotensin-aldosterone systems, including hemodynamic changes, such as increased preload, peripheral vasodilatation, positive inotropic and chronotropic effects. Activation of the sympathetic and renin-angiotensin-aldosterone system may also have a negative effect on cardiac function through cardiac fibrosis, myocardial cell death, left ventricular hypertrophy and dilatation, which can lead to a decreased left ventricular ejection fraction [9,10] and related high HF biomarkers (such as natriuretic peptides or troponin levels) [11]. The stretching of cardiac myocytes possibly leads to a leakage of the cytosolic pool of troponin T by the transient loss of cell membrane integrity [12]. Therefore, cardiac troponin T detected in plasma is a highly specific marker of myocardial damage (necrosis). High-sensitivity troponin tests, available for the past several years, detect troponin levels with a high degree of credibility [11]. Moreover, anemia is often associated with iron deficiency and renal insufficiency, conditions which can also worsen HF. Consequently, the presence of anemia alone is also associated with poor outcomes in all types of HF, regardless of the left ventricular ejection fraction (LVEF) [6,7,13,14,15].

At the same time, HF can cause anemia due to systemic venous congestion and related hemodilution or an absorption deficiency (e.g., B12, folic acid, iron), including the elevation of cytokines and inflammatory factors, bone marrow deficiency and renal dysfunction [6,7,8].

Nevertheless, there are limited data on the relevance of anemia on outcomes in patients with over 40% of LVEF in such subgroups as HFpEF, HFmrEF and the newly defined HF group with recovered EF (HFrecEF).

Heart failure subclassification by LVEF was changed in 2021, and separately defined those HF patients with reduced EF (HFrEF), in whom EF had improved and reached at least 40% due to the treatment [16]. Their outcome and prognosis may be better, than HFrEF patients, although limited data are available about the comparison with those HFpEF-HFmrEF cohorts, in whom LVEF is similar.

Therefore, the aim of this study was to describe the characterization of our patients with HFpEF or HFmrEF and assess their outcome by the presence of anemia using our single-center retrospective database; moreover, the objective was to compare them with those with HFrecEF [previously HFrEF patients after cardiac resynchronization therapy (CRT) implantation] who also have >40% LVEF with and without anemia.

## 2. Methods

### 2.1. Patient Population

#### 2.1.1. HFpEF-HFmrEF Cohort

Patients with a diagnosis of HFpEF or HFmrEF at Semmelweis University Heart and Vascular Center between April 2013 and September 2019 were registered for this study. Based on the current guidelines [17], only those who had a baseline LVEF ≥ 40% and echocardiographic evidence of structural abnormalities as left ventricular hypertrophy and/or left atrial enlargement and/or diastolic dysfunction were enrolled. Additionally, patients had elevated serum NT-proBNP level (in sinus rhythm > 300 pg/mL, in atrial fibrillation >500 pg/mL) and showed HF symptoms at baseline (NYHA functional class II-IVa). Those with LVEF < 40% at any time before the inclusion, or if there was an acute event at the time of inclusion or within three months (e.g., myocardial infarction, electrical cardioversion, major bleeding and other interventions) were excluded.

#### 2.1.2. HFrecEF Cohort

We also selected an HFrecEF cohort from our retrospective cardiac resynchronization therapy (CRT) database, which comprised 2524 patients who underwent CRT implantation in our center between 2005 and 2018, with patients’ characteristics that were previously published in detail [18,19,20,21]. Patients were considered for device implantation as per the current guidelines: symptomatic (NYHA II-IVa) HF patients with reduced EF (LVEF < 35%), and wide QRS (>130 ms) despite the optimal medical treatment. From this cohort, 69% had LBBB morphology, showing a high rate of class I for implantation.

First, patients in whom LVEF improved to over 40% after 12 months assessed by echocardiography (meaning at least a 5% absolute increase in their LVEF), were considered as responders.

Left ventricular ejection fraction parameters measured by the two-dimensional biplane Simpson method were registered. Measurements were performed following our institutional protocol. Based on our quality assurance and experience, the interobserver variability rate was similarly low as in academic centers in Europe.

Altogether, there were 138 HFrecEF patients who proved to be responders and served as a pool for propensity score matching (PSM), in which 75 individuals were finally selected. These patients were matched by their LVEF, serum creatinine levels, etiology, gender and age with 75 HFpEF-HFmrEF patients.

### 2.2. Baseline Evaluation, Endpoint and Follow-Up

We recorded the patients’ medical history, physical status, echocardiographic-, ECG-, laboratory parameters and their medical therapy. Anemia was defined according to WHO criteria as a serum hemoglobin (Hbg) level < 12 g/dL in women and < 13 g/dL in men.

Patients’ data were exported and analyzed anonymously. Patients’ status (dead or alive) and the date of death were obtained by querying the National Health Insurance Database of Hungary.

The study protocol complies with the Declaration of Helsinki, and the protocol was approved by the Medical Research Council (ETT TUKEB No. 161/2019).

The primary endpoint was defined as death from any cause.

### 2.3. Statistics

In the description of baseline clinical characteristics, in the case of normal distribution, continuous variables are expressed as means and standard deviations (SDs), whereas non-normally distributed data are expressed as medians and interquartile ranges (IQRs). To determine whether the data are distributed normally, we performed Shapiro–Wilk tests. Categorical variables are described with numbers and percentages.

To compare continuous variables within the same group, paired Student’s *t*-tests or paired Wilcoxon rank tests were performed, as appropriate. Depending on normality, continuous variables between groups were compared using unpaired Student’s *t*-test or Mann–Whitney U tests. To compare categorical variables, chi-squared or Fisher’s exact tests were performed. Time-to-event analyses were performed using log-rank tests, univariable and multivariable Cox regression analyses. The differences between groups in the case of *p* < 0.05 values were considered significant. Statistical analyses were performed using GraphPad Prism (version 8, Inc., GraphPad Software, San Diego, CA, USA), MedCalc (version 19.6.4, MedCalc Software Ltd., Ostend, Belgium), and SPSS Statistics (version 25.0, IBM, Armonk, NY, USA) programs. We performed propensity score matching in R (version 3.6.3, R Foundation for Statistical Computing, Vienna, Austria) using the MatchIt package (version 3.0.2). After replacing the missing values with the mean of the non-missing cases, propensity score matching was performed using the nearest neighbor matching (distance calculated with the logistic regression method), selecting 75 patients each from HFpEF-HFmrEF and HFrecEF cohorts, respectively. During selection LVEF, serum creatinine levels, etiology, gender and age were matched.

## 3. Results

### 3.1. Baseline Clinical Characteristics of the Total Patient Cohort

Altogether, 375 patients were analyzed in the HFpEF-HFmrEF patient group. The median age of the cohort was 75 (IQR 69–82) years old, with a median LVEF of 55% (IQR 5060). Over half of the patients were female (52%), 57% had ischemic etiology, and 38% had severe symptoms (NYHA III or IVa functional class) (Table 1).

Regarding comorbidities, 91% of the total cohort had hypertension, 63% had atrial fibrillation and approximately 50% of them had coronary artery disease or chronic kidney disease; 138 patients (37%) had anemia, 36% had type 2 diabetes mellitus, and 16% had COPD (Figure 1).

### 3.2. Baseline Clinical Characteristics by Ejection Fraction

From the total cohort, 326 (87%) patients had HFpEF and 49 (13%) had HFmrEF. In the HFmrEF group there were significantly more male patients (67% vs. 45%, *p* < 0.01), with ischemic etiology (74% vs. 55%, *p* = 0.01), coronary artery disease (67% vs. 49%, *p* = 0.02) and myocardial infarction (MI) (51% vs. 29%, *p* < 0.01), also more frequently compared with the HFpEF group. In the HFpEF group, the mean serum NT-proBNP level was lower [1070 (IQR 603–1887) pg/mL vs. 1543 (IQR 684–28,160 pg/mL; *p* = 0.02], and significantly more patients were treated by an ACE inhibitor or ARB (86% vs. 74%, *p* = 0.03), but fewer received furosemide (FSD) therapy (60% vs. 76%, *p* = 0.04) (Table 1).

### 3.3. Baseline Clinical Characteristics by the Presence of Anemia

Patients were also dichotomized by the presence of anemia. In the total cohort, 138 (37%) patients had anemia (Table 2), 38% showed mildly reduced serum hemoglobin (Hbg < 11 g/dL), 14 (10%) patients had moderately reduced values (<9.2 g/dL), and only 8 (6%) had severe anemia (<8 g/dL). In patients with anemia, the proportion of men was higher (56% vs. 43%, *p* = 0.01), and more patients had severe symptoms (NYHA III/IV functional class: 53% vs. 28%, *p* < 0.01) and had lower diastolic blood pressure (75 ± 13 mmHg vs. 82 ± 12 mmHg, *p* < 0.01) compared with those without anemia.

Regarding comorbidities, subjects with decreased Hgb levels had type 2 diabetes (43% vs. 32%, *p* = 0.03), chronic kidney disease (63% vs. 44%, *p* < 0.01) or CABG (15% vs. 6%, *p* < 0.01) more frequently in their medical history.

Laboratory parameters also reflected such differences; patients with anemia had higher serum creatinine [105 (IQR 79–136) µmol/L vs. 87 (IQR 75–105) µmol/L, *p* < 0.01], urea—[8.2 (IQR 6.3–10.8) mmol/L vs. 6.7 (IQR 5.4–8.4) mmol/L, *p* < 0.0], and NT-proBNP levels [1584 (IQR 761–3088) pg/mL vs. 912 (IQR 558–1647) pg/mL, *p* < 0.01], and their eGFR values [52 (IQR 38–67) mL/min/1.73 m2 vs. 61 (IQR 50–79) mL/min/1.73 m2, *p* < 0.01] were also lower than in patients with no anemia.

Medical treatment also highlighted these differences, more patients with low serum Hgb received furosemide (72% vs. 56%, *p* < 0.01), but fewer tolerated beta-blocker therapy (81% vs. 90%, *p* = 0.02) compared with those without the presence of anemia. Regarding oral anticoagulant therapy, a similar proportion of patients received this treatment. In the anemia group 51% were added, mainly receiving direct oral anticoagulation (DOAC) therapy, whereas in patients with a normal Hgb level, 62% received anticoagulation, out of which 60% received DOAC (51% vs. 62%; *p* = 0.10).

### 3.4. Outcome of the HFpEF-HFmrEF Patient Cohort

In our study, 42 (11%) patients died in the total patient population during the median follow-up time of 1.4 (IQR 1–1.8) years. In the univariate model, anemia (HR 2.77, 95% CI 1.47–5.23, *p* < 0.01), and furosemide therapy (HR 3.02, 95% CI 1.34–6.80, *p* < 0.01) were proved to be the most relevant predictors of all-cause mortality. Altogether, 25 (18%) patients in the anemia group and 17 (7%) patients without anemia reached the primary endpoint (HR 2.77, 95% CI 1.47–5.23, *p* < 0.01) (Figure 2). The prognostic role of anemia on death from any cause was also confirmed by Cox regression analysis after adjusting for age, gender, serum creatinine level and furosemide intake (aHR 2.33, 95% CI 1.21–4.52, *p* = 0.01) (Figure 2).

In a subgroup analysis by gender, the risk of all-cause mortality in patients with anemia was higher in males and females, respectively, but the difference was more pronounced in men (HR 3.08, 95% CI 1.35–7.05, *p* < 0.01) (Appendix A) compared with women (HR 2.59, 95% CI 0.93–7.26, *p* = 0.03) (Appendix A).

### 3.5. Baseline Clinical Characteristics of the Selected HFpEF-HFmrEF and HFrecEF Cohort by Propensity Score Matching

Altogether, 75 patients from each cohort were selected by propensity score matching (PSM) analysis with comparable values of LVEF, serum creatinine levels, etiology, gender and age.

The median age of HFpEF-HFmrEF patients vs. HFrecEF patients [71.5 (IQR 64–79) vs. 70 (IQR 63–74); *p* = 0.09] was almost 70 years old, and the majority of the patients were male (67% vs. 64%; *p* = 0.73). The basic anthropometric measures, LVEF, and laboratory parameters, such as serum hemoglobin levels [13.5 (IQR 12–14) g/dL vs. 13 (IQR 2–15) g/dL, *p* = 0.72] or serum creatinine levels [93 (IQR 77–117) µmol/L vs. 96 (IQR 73–122) µmol/L, *p* = 0.83], were also comparable between the two groups. Regarding comorbidities, hypertension (87% vs. 75%, *p* = 0.06), ischemic etiology (53% vs. 48%, *p* = 0.51), chronic kidney disease (43% vs. 48%, *p* = 0.51), COPD (19% vs. 11%, *p* = 0.17), and diabetes (39% vs. 37%, *p* = 0.82) were the most prevalent (Table 3).

### 3.6. Outcome of the HFpEF-HFmrEF vs. HFrecEF Patient Cohort

Comparing the primary endpoint in the HFpEF-HFmrEF and HFrecEF cohorts, there was no significant difference between the two patient groups (HFrecEF HR 1.22, 95% CI 0.52–2.89, *p* = 0.59) (Appendix A).

Both groups were dichotomized by the presence of anemia. The proportion of patients with decreased Hgb level was 22% in the HFpEF-HFmrEF group, and 24% in the HFrecEF group. The prognosis of patients with anemia was significantly worse in both groups: in the HFpEF-HFmrEF group the risk of all-cause mortality was more than four times higher (HR 4.36, 95% CI 0.93–20.50, *p* = 0.03); in the HFrecEF group, it was almost ten times higher (HR 9.64, 95% CI 3.44–27.00, *p* < 0.01) (Figure 3). However, there was no significant difference (HR 1.42, 95% CI 0.47–4.35, *p* = 0.45) between the two types of heart failure (HFpEF-HFmrEF vs. HFrecEF) in terms of the prevalence of anemia (Table 4).

## 4. Discussion

The main findings of our study can be summarized as follows:Our HFpEF-HFmrEF cohort is a vulnerable group with a high frequency of comorbidities;Although the proportion of HFmrEF patients was low in our patient cohort, the characteristics of these patients differed from those with HFpEF: in terms of the ischemic etiology, the proportions of men and serum NT-proBNP levels were higher;More than one-third of our patients suffered from anemia, and exhibited advanced heart failure symptoms, laboratory- and echocardiographic parameters;In the total patient cohort, in addition to furosemide therapy, anemia was an independent predictor of all-cause mortality, and the risk of death was almost three times higher than in those with normal hemoglobin levels;The presence of anemia was associated with a significantly higher risk of all-cause mortality in HFpEF-HFmrEF and those with >40% of LVEF from HFrecEF patients compared with those without anemia;By propensity score matching, the outcomes of HFpEF-HFmrEF and HFrecEF patients with anemia were poor and did not differ significantly.

The prevalence of HFpEF or HFmrEF is high, reaching up to 3–5% in the average population, which will further increase in the future due to aging and the ever-increasing prevalence of comorbidities [22,23,24,25,26,27,28,29]. The frequency of comorbidities is high in this population, and until 2021 there was no known effective treatment to improve the mortality of this patient cohort [30], in comparison with patients with HFrEF [31,32]. In 2021, the EMPEROR-Preserved trial showed that empagliflozin reduced the risk of cardiovascular death or hospitalization for HF in HFpEF patients.

Nevertheless, the subgroup of heart failure patients >40% LVEF was reclassified in 2021, extending the HFmrEF and HFpEF groups with a third category [16]; those patients who previously had a reduced EF (HFrEF), but improved over 40% due to the treatment (HFimpEF/HFrecEF) [16]. Limited data are available about their outcomes compared with HFpEF and HFmrEF patients; therefore, we also performed further analyses in this regard.

Therefore, firstly, we aimed to describe the outcomes of our HFpEF or HFmrEF cohort by the presence of anemia, and secondly, we aimed to compare them with those with HFrecEF patients without anemia.

In this study, as we selected our patients based on the criteria of recent HFpEF trials [30,33]; the baseline clinical characteristics of our patients were very similar to those presented in the latest trials [6,15]. The mean LVEF was 55%, which was almost identical to the mean values described in the EMPEROR-Preserved (54%) and PARAGON (57%) trials [30,34]. The mean age was higher in HFpEF patients than those with HFrEF, and our cohort was even older than participants in the EMPEROR-Preserved trial (75 years vs. 72 years) [30]. Regarding the etiology, we found a large difference between our cohort and previous study groups, because in the latter, nearly one-third of the enrolled subjects had an ischemic etiology, which reached 57% in our group, and the presence of myocardial infarction or coronary intervention was almost two times higher in our patients’ medical history [30,34]. This phenomenon cannot be explained by the rate of HFmrEF patients in our study, which was considerably lower (15%) as compared with the EMPEROR-Preserved study (30%) [20]. However, it is important to note that in our research, the characteristics of the HFmrEF group differed from those of HFpEF, because the ischemic etiology, the proportion of men, and the NT-proBNP levels were higher.

It is well known that the number of comorbidities proportionally increases the incidence of HFpEF [6]. Thus, the HFpEF population is very heterogeneous, and their outcomes are significantly influenced by comorbidities [6,35]. The presence and rates of different comorbidities in our research were similar to those in the study by Lund: hypertension (91% vs. 78%), atrial fibrillation (63% vs. 65%), coronary heart disease (52% vs. 33%), chronic renal failure (51% vs. 46%), anemia (37% vs. 51%) and diabetes mellitus (36% vs. 30%) occurred most commonly [6].

The prevalence of anemia in HFpEF patients is described in a wide range (19–68%) [13,35]. In our study population, 37% of the patients had anemia, of which 57% were male. These results are identical to those of previous studies focusing on the prevalence of anemia in this population [15]. Regarding their symptoms, more than half (53%) of our patients in the anemia group had an advanced NYHA class (III/IV), which was even higher than in the MAGGIC study (53% vs. 36%) [13]. Serum NT-proBNP levels also confirmed that this cohort had an advanced heart failure with a mean value of 1500 pg/mL, which was significantly higher than described in large-scale RCTs [34]. This can explain the difference in medication, because more patients in the anemia group received furosemide (72% vs. 56%) and fewer beta-blockers (81% vs. 90%) than in the non-anemia group. Since patients with anemia tend to have a worse NYHA class, they are more likely to receive furosemide therapy to reduce the congestion and related symptoms. However, adding diuretics in anemic patients results in a reduction in oxygen delivery, which causes a vicious cycle.

In the HF population, several pathomechanisms (e.g., congestion and related malabsorption, renal dysfunction and impaired erythropoietin production, general inflammation and frequent use of anticoagulants) are responsible for the development of anemia. Anemia can be related to hemodilution due to fluid retention, which is a classic clinical feature of patients presenting with HF. Malabsorption-related deficiencies of hematinic vitamins, such as B12 or folate are not so common in anemic patients with HF, while anorexia, malabsorption and aspirin-induced gastrointestinal bleeding is the main cause of iron deficiency anemia in the elderly population. Erythropoietin, which stimulates red blood cell production is often abnormal in HF patients. Although renal dysfunction is frequent in HF, such structural renal diseases that could reduce EPO production are scarce. During HF, renal blood flow is decreased which should increase EPO production. While EPO levels are indeed increased, they are still lower than expected for the degree of anemia, suggesting blunted EPO production. Nevertheless, the relationship between renal blood flow and EPO secretion in HF is complex and not completely understood yet. Inflammation is an important component of HF with elevated C-reactive protein (CRP) and proinflammatory cytokine levels, such as interleukin-6 (IL-6) and tumor necrosis factor-α (TNF-α). IL-6 and TNF-α inhibit renal erythropoietin production in the kidney by activating GATA binding protein 2 and nuclear factor- κB, which may explain the blunted erythropoietin response in HF. These proinflammatory cytokines also inhibit the proliferation of bone marrow erythroid progenitor cells. Anticoagulant therapy used in atrial fibrillation, which is one of the most frequent comorbidities in HF, can also cause bleedings which possibly leads to anemia. Thus, a large proportion of patients suffers from low hemoglobin levels and/or an iron deficiency across the entire spectrum of LVEF, negatively influencing the patients’ outcome [6,7,13,14,15,36,37]. Using data from the Swedish HF Registry of nearly 50,000 patients, Lund investigated the predictive role of anemia on long-term all-cause mortality in the HFrEF, HFmrEF, and HFpEF populations. [6]. Their results showed that the incidence of anemia was higher with increases in the LVEF, findings in line with ours, showing that the presence of anemia increased the risk of all-cause mortality in all groups [6].

In our research, it should be emphasized that the outcome was also assessed by LVEF, but in a unique way. HFrecEF is a recently classified group of heart failure patients; therefore, there are limited data on their survival directly compared with HFmrEF-HFpEF patients. According to our results, systolic heart failure patients who are expecting the most beneficial outcome after a positive response to CRT have similar medium-term all-cause mortality risks to HFpEF-HFmrEF patients [18,19,20,21]. In addition, the presence of anemia significantly worsened the outcome of patients regardless of heart failure etiology. In the HFpEF–HFmrEF group, patients with anemia had a more than four times higher all-cause mortality risk; in the HFrecEF group, there was a nearly ten times higher risk compared with patients with normal serum hemoglobin levels. This proportion is much higher than that described by Lund [35]. These results also highlight the importance of describing the characteristics and the outcome of different heart failure patients by site, region or country, since HF affects mostly the older population with multiple comorbidities, major differences might be revealed by region.

Altogether, anemia has been shown to be an independent predictor of all-cause mortality in HFpEF and HFmrEF patients; moreover, it has a negative influence on outcomes of over 40% of the left ventricular ejection fraction.

There are a few limitations to be acknowledged. Our research was a single-center, retrospective analysis with all of such databases’ disadvantages, like the difficulty of assessing temporal relationships, or the inferior level of evidence compared with prospective studies. We cannot control the exposure assessment, instead we must rely on others for accurate recordkeeping. Although the diagnoses of HFpEF and HFmrEF were based on LVEF, structural abnormalities and serum NT-proBNP levels, some echocardiographic parameters, determined by Tissue Doppler imaging were often missing. To exclude patients with anemia due to other etiologies, we considered the history of active bleeding; however, B12, folic acid levels, parameters for characterizing the iron state (as iron level, ferritin, ferritin saturation, transferrin) and Weber tests for objectifying the occult bleeding were not performed. Furthermore, gastrointestinal diseases such as Crohn’s disease, ulcerative colitis, or coeliac disease were not recorded. Control serum Hgb levels were not taken; thus, the chronic existence of anemia is uncertain.

## 5. Conclusions

In our retrospective, single-center study, we found that the group of heart failure patients with a preserved and mildly reduced ejection fraction was an older, vulnerable cohort with a high frequency of comorbidities. Among comorbidities, anemia proved to be the most relevant, affecting almost one third of patients, with a three times higher all-cause mortality risk, and could be considered as an independent predictor of mortality. Moreover, regardless of the type of heart failure over 40% of the ejection fraction (HFpEF-HFmrEF/HFrecEF), the presence of anemia impaired the clinical outcome; thus, it highlighted the importance of early diagnostics and treatment of anemia in HF patients.

## Figures and Tables

**Figure 1 diagnostics-12-00517-f001:**
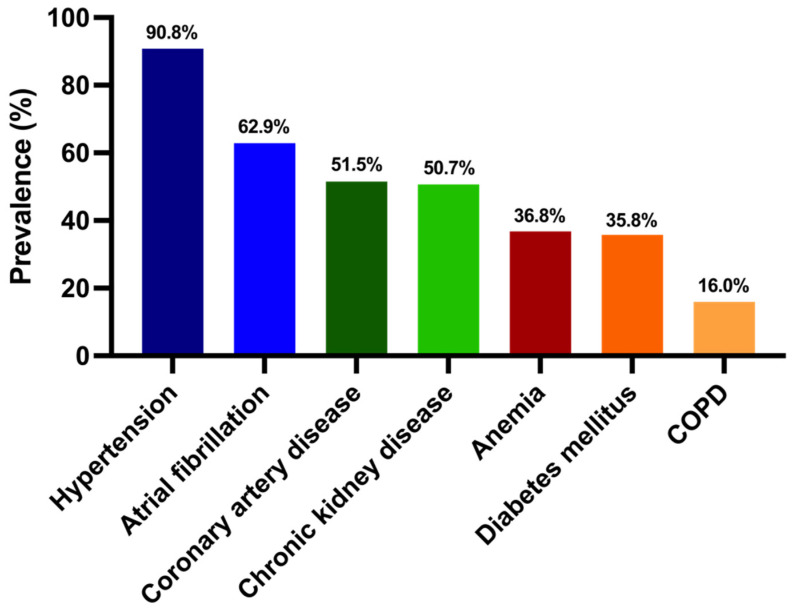
Prevalence of comorbidities in our cohort.

**Figure 2 diagnostics-12-00517-f002:**
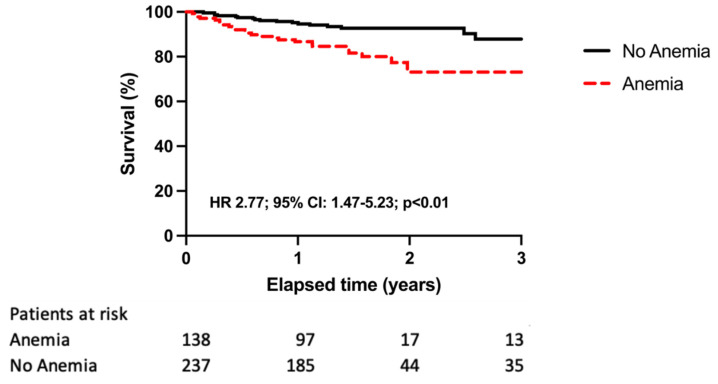
Kaplan–Meier estimates of the probability of survival according to the presence vs. absence of anemia.

**Figure 3 diagnostics-12-00517-f003:**
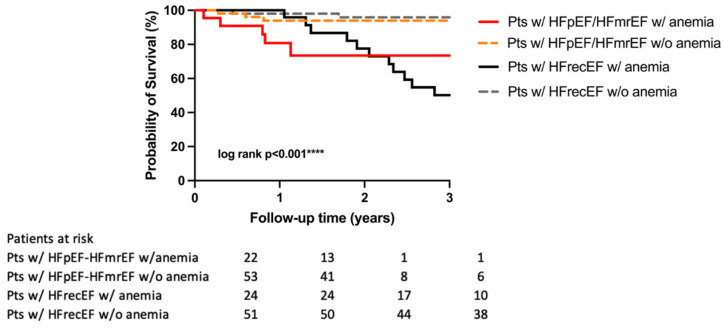
Kaplan–Meier estimates of the probability of survival in HFpEF–HFmrEF patients with and without anemia, HFrecEF patients with low and with normal serum hemoglobin levels.

**Table 1 diagnostics-12-00517-t001:** Baseline clinical characteristics of heart failure patients with preserved (HFpEF) and with mildly reduced (HFmrEF) ejection fractions.

Baseline Variables	All Patients (*n* = 375)	HFpEF (*n* = 326)	HFmrEF (*n* = 49)	*p* Value
Age (years, median/IQR)	75 (69–82)	76 (69–82)	74 (68–80)	0.21
Gender (female, *n*, %)	196 (52)	180 (55)	16 (33)	<0.01 ***
NYHA III/IV (*n*, %)	129 (38)	109 (37)	20 (42)	0.64
Ischemic etiology (*n*, %)	212 (57)	176 (55)	36 (74)	0.01 **
Systolic BP (mmHg, median/IQR)	135 (123–151)	136 (125–152)	131 (118–140)	0.06
Diastolic BP (mmHg, mean ± SD)	79 ± 13	79 ± 13	80 ± 10	0.86
BMI (kg/m^2^, median/IQR)	30.1 (25.9–34.1)	30.1 (26.0–33.9)	30.0 (25.4–36.7)	0.80
LBBB morphology (*n*, %)	25 (7)	22 (7)	3 (7)	1.0
Medical history				
Hypertension (*n*, %)	334 (91)	293 (91)	41 (87)	0.42
Atrial fibrillation (*n*, %)	236 (63)	206 (63)	30 (61)	0.87
CAD (*n*, %)	193 (52)	160 (49)	33 (67)	0.02 *
CKD (*n*, %)	190 (51)	169 (52)	21 (43)	0.28
T2DM (*n*, %)	133 (36)	116 (36)	17 (35)	1.00
Prior MI (*n*, %)	118 (32)	93 (29)	25 (51)	<0.01 ***
Prior PCI (*n*, %)	123 (33)	103 (32)	20 (41)	0.25
Prior CABG (*n*, %)	35 (9)	27 (8)	8 (17)	0.11
COPD (*n*, %)	60 (16)	52 (16)	8 (16)	1.00
Laboratory parameters				
Creatinine (µmol/L, median/IQR)	92 (76–117)	91 (76–117)	93 (81–116)	0.83
Cholesterol (mmol/L, median/IQR)	4.2 (3.4–5.0)	4.2 (3.4–5.0)	3.8 (3.1–4.6)	0.12
eGFR (ml/min/1.73 m^2^, median/IQR)	58 (46–78)	58 (45–77)	67 (50–81)	0.07
Urea (mmol/L, median/IQR)	7.2 (5.7–9.0)	7.2 (5.7–9.0)	7.2 (5.0–9.0)	0.26
NT-proBNP (pg/mL, median/IQR)	1085 (621–1973)	1070 (603–1887)	1543 (684–2816)	0.02 *
Echocardiographic parameters				
LVEF (%, median/IQR)	55 (50–60)	56 (55–60)	45 (42–47)	NA
LVEDD (mm, median/IQR)	47 (43–51)	46 (42–51)	50 (46–53)	<0.01 ***
LVESD (mm, mean/IQR)	32 (28–37)	31 (27–36)	37 (34–43)	<0.01 ***
Medical treatment				
Beta blocker (*n*, %)	321 (87)	276 (85)	45 (94)	0.17
ACEi/ARB (*n*, %)	315 (85)	279 (86)	36 (74)	0.03 *
MRA (*n*, %)	105 (28)	88 (27)	17 (35)	0.31
Furosemide (*n*, %)	231 (62)	194 (60)	37 (75)	0.04 *
OAC (*n*, %)	217 (59)	191 (60)	26 (53)	0.44

ACEi, angiotensin-converting-enzyme inhibitor; ARB, angiotensin receptor blocker; BMI, body mass index; CABG, coronary artery bypass grafting; COPD, chronic obstructive pulmonary disease; CAD, coronary artery disease; CKD, chronic kidney disease; T2DM, type 2 diabetes mellitus; eGFR, estimated glomerular filtration rate; LBBB, left bundle branch block; LVEDD, left ventricular end-diastolic diameter; LVEF, left ventricular ejection fraction; LVESD, left ventricular end-systolic diameter; MI, myocardial infarction; MRA, mineralocorticoid receptor antagonist; NT-proBNP, N-terminal pro-B-type natriuretic peptide; NYHA, New York Heart Association stage; OAC, oral anticoagulant; PCI, percutaneous coronary intervention. * *p* < 0.05, ** *p* < 0.01, *** *p* < 0.001.

**Table 2 diagnostics-12-00517-t002:** Baseline clinical characteristics of patients without anemia.

Baseline Variables	All Patients (*n* = 375)	With Anemia (*n* = 138)	Without Anemia (*n* = 237)	*p* Value
Age (years; median/IQR)	75 (69–82)	77 (70–82)	75 (69–81)	0.08
Gender (female; *n*, %)	196 (52)	60 (44)	136 (57)	0.01 **
NYHA III/IV (*n*, %)	129 (38)	70 (53)	59 (28)	<0.01 ***
Ischemic etiology (*n*, %)	212 (57)	83 (61)	129 (55)	0.31
Systolic BP (mmHg, median/IQR)	135 (123–151)	133 (124–154)	137 (122–150)	0.94
Diastolic BP (mmHg, mean ± SD)	79 ± 13	75 ± 13	82 ± 12	<0.01 ***
BMI (kg/m^2^, median/IQR)	30.1 (25.9–34.1)	29.7 (25.4–33.6)	30.2 (26.2–34.5)	0.37
LBBB morphology (*n*, %)	25 (7)	11 (8)	15 (7)	0.67
Medical history				
Hypertension (*n*, %)	334 (91)	125 (92)	209 (90)	0.71
Atrial fibrillation (*n*, %)	236 (63)	81 (59)	155 (65)	0.22
CAD (*n*, %)	193 (52)	75 (54)	118 (50)	0.45
CKD (*n*, %)	190 (51)	87 (63)	103 (44)	<0.01 ***
T2DM (*n*, %)	133 (36)	59 (43)	74 (32)	0.03 *
Prior MI (*n*, %)	118 (32)	48 (35)	70 (30)	0.30
Prior PCI (*n*, %)	123 (33)	48 (35)	75 (32)	0.57
Prior CABG (*n*, %)	35 (9)	21 (15)	14 (6)	<0.01 ***
COPD (*n*, %)	60 (16)	26 (19)	34 (14)	0.31
Laboratory parameters				
Creatinine (µmol/L, median/IQR)	92 (76–117)	105 (79–136)	87 (75–105)	<0.01 ***
Cholesterol (mmol/L, median/IQR)	4.2 (3.4–5.0)	3.8 (3.1–4.5)	4.4 (3.7–5.3)	<0.01 ***
eGFR (ml/min/1.73 m^2^, median/IQR)	58 (46–78)	52 (38–67)	61 (50–79)	<0.01 ***
Urea (mmol/L, median/IQR)	7.2 (5.7–9.0)	8.2 (6.3–10.8)	6.7 (5.4–8.4)	<0.01 ***
NT-proBNP (pg/mL, median/IQR)	1085 (621–1973)	1584 (761–3088)	912 (558–1647)	<0.01 ***
Echocardiographic parameters				
LVEF (%, median/IQR)	55 (50–60)	55 (52–60)	55 (50–60)	0.81
LVEDD (mm, median/IQR)	47 (43–51)	47 (44–51)	46 (41–51)	0.04 *
LVESD (mm, mean/IQR)	32 (28–37)	33 (29–38)	31 (27–36)	0.02 *
Medical treatment				
Beta blocker (*n*, %)	321 (87)	109 (81)	212 (90)	0.02 *
ACEi/ARB (*n*, %)	315 (85)	113 (83)	202 (85)	0.56
MRA (*n*, %)	105 (28)	38 (28)	67 (28)	1.00
Furosemide (*n*, %)	231 (62)	98 (72)	133 (56)	<0.01 ***
OAC (*n*, %)	217 (59)	71 (53)	146 (62)	0.10

ACEi, angiotensin-converting-enzyme inhibitor; ARB, angiotensin receptor blocker; BMI, body mass index; CABG, coronary artery bypass grafting; COPD, chronic obstructive pulmonary disease; CAD, coronary artery disease; CKD, chronic kidney disease; T2DM, type 2 diabetes mellitus; eGFR, estimated glomerular filtration rate; LBBB, left bundle branch block; LVEDD, left ventricular end-diastolic diameter; LVEF, left ventricular ejection fraction; LVESD, left ventricular end-systolic diameter; MI, myocardial infarction; MRA, mineralocorticoid receptor antagonist; NT-proBNP, N-terminal pro-B-type natriuretic peptide; NYHA, New York Heart Association stage; OAC, oral anticoagulant; PCI, percutaneous coronary intervention. * *p* < 0.05, ** *p* < 0.01, *** *p* < 0.001.

**Table 3 diagnostics-12-00517-t003:** Baseline clinical characteristics of HFpEF-HFmrEF and HFrecEF groups.

Baseline Variables	HFpEF-HFmrEF Patients (*n* = 75)	HFrecEF Patients (*n* = 75)	*p* Value
Age (years, median/IQR)	71.5 (63.7–79.4)	70.2 (62.8–74.4)	0.09
Gender (female, *n*, %)	25 (33)	27 (36)	0.73
Ischemic etiology (*n*, %)	40 (53)	36 (48)	0.51
Nonischemic etiology (*n*, %)	35 (47)	39 (52)	0.51
Systolic BP (mmHg, median/IQR)	132 (124–145)	128 (114–145)	0.32
Heart rate (1/min, median/IQR)	76 (65–85)	75 (75–75)	0.90
Weight (kg, median/IQR)	85 (80–103)	80 (73–96)	0.07
Height (cm, median/IQR)	172 (162–178)	170 (162–176)	0.84
BMI (kg/m^2^, median/IQR)	31 (27–35)	28 (26–34)	0.06
Medical history			
Hypertension (*n*, %)	65 (87)	56 (75)	0.06
CKD (*n*, %)	32 (43)	36 (48)	0.51
T2DM (*n*, %)	29 (39)	28 (37)	0.82
Prior MI (*n*, %)	31 (41)	27 (36)	0.50
Prior PCI (*n*, %)	47 (63)	49 (66)	0.65
Anemia (*n*, %)	22 (29)	24 (32)	0.72
Nicotinismus (*n*, %)	2 (3)	2 (3)	0.99
COPD (*n*, %)	15 (19)	8 (11)	0.1
Laboratory parameters			
Creatinine (µmol/L, median/IQR)	93 (77–117)	96 (73–122)	0.83
Hemoglobin (g/dL, median/IQR)	13.5 (12.1–14.4)	13.1 (12–14.6)	0.72
eGFR (ml/min/1.73 m^2^, median/IQR)	62 (48–83)	60 (47–83)	0.50
Echocardiographic parameter			
LVEF (%, median/IQR))	48 (45–50)	45 (41–50)	0.06

BMI, body mass index; COPD, chronic obstructive pulmonary disease; CKD, chronic kidney disease; T2DM, type 2 diabetes mellitus; eGFR, estimated glomerular filtration rate; LVEF, left ventricular ejection fraction; MI, myocardial infarction; PCI, percutaneous coronary intervention.

**Table 4 diagnostics-12-00517-t004:** All-cause mortality risk based on the type of heart failure and on the presence of anemia.

All-Cause Mortality	HR	95% CI	*p* Value
HFpEF-HFmrEF patients with anemia vs. HFpEF-HFmrEF patients with normal serum hemoglobin levels	4.36	0.93/20.50	0.03 *
HFrecEF patients with anemia vs. HFrecEF patients with normal serum hemoglobin levels	9.64	3.44/27.00	<0.01 ***
HFpEF-HFmrEF patients with anemia vs. HFrecEF patients with anemia	1.42	0.47/4.35	0.45
HFpEF-HFmrEF patients without anemia vs. HFrecEF patients without anemia	1.55	0.33/7.24	0.48

* *p* < 0.05, *** *p* < 0.001.

## Data Availability

Not applicable.

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
