# Peer review of "The Prognostic Value of Anemia in Patients with Preserved, Mildly Reduced and Recovered Ejection Fraction"

_diagnostics, 2022, doi:10.3390/diagnostics12020517_

Round 1

Reviewer 1 Report

In this work, the authors underwent to show how the presence of anemia in the population of patients with mildly reduced (HFmrEF) and preserved (HFpEF) ejection fraction modifies the primary outcome of all-cause death during the follow-up period. The work is of some interest and anemia is an important therapeutic target in this patient population. However, I would have some comments, as outlined below:

  1. I am interested to know how did you diagnose HFpEF and what diagnostic algorithm did you use to classify your patients as HFpEF? This should be disclosed and explained in the detail. As authors are aware, HFpEF is often a difficult diagnosis and should be backed up by the evidence of increased filling pressures, exercise intolerance, and by using diagnostic algorithm, for example, HFA-PEFF score from HFA. This crucial detail should be elaborated.
  2. It would be beneficial if the authors would be able to delineate types of anemia and also an iron deficiency in this cohort? What type of anemia was this...not just defined by hemoglobin levels? How many patients had iron-deficiency anemia, or anemia of chronic disease? This should be discussed or if not possible, at least acknowledged as a major limitation.
  3. How many people in your cohort used ARNI or were administered intravenous iron supplements?
  4. Authors should incorporate and acknowledge some limitations of their work in the manuscript structure.
  5. Authors fail to characterize and put forward details on what they exactly mean by saying HFrecEF population? Who were these patients besides the fact that they underwent CRT implantation? What did the authors use as a criterion to classify someone as "recovered" with respect to systolic function? This should be stated more clearly. It is most certainly likely that these patients were HFrEF, had LBBB ECG morphology, and QRS wider than 130 or 150 msec. This should be clearly stated. 
  6. To me, it remains unclear what are the exact main purposes of this study. The authors state in the title that they describe the population with HFmrEF and HFpEF with respect to the presence of anemia, however, in the manuscript they go on by incorporating the HFrecEF population and this is not even mentioned in the title. This might seem misleading, therefore, I suggest authors make certain rearrangements in the text and also the manuscript title to better reflect what was actually done.
  7. Please remove the word "Se" from the tables indicating serum. For example, Se creatinine, Se cholesterol, etc. This is redundant.
  8. Correct "hypentension" to "hypertension" in Table 1.
  9. Please report the median duration of your follow-up with interquartile ranges rather than mean and standard deviation.
  10. How was LVEF measured? Was this one measurement or many measurements? A single operator or many operators? Which technique was used to measure LVEF?
  11. The manuscript lacks many important technical details on the procedures and methods used. Please do general rewriting of this portion.
  12. Since the main endpoint of the interest was all-cause mortality, authors should show that their regression analysis was multivariable-adjusted for other competing risks and confounding variables that could contribute to all-cause death. This is not done and the regression model is poorly explained in the manuscript.

Reviewer 2 Report

Thank you for the opportunity to review your manuscript entitled "The Prognostic Value of Anemia in Patients with Preserved and Mildly Reduced Ejection Fraction".

Anaemia is defined by the Word Health Organisation as a haemoglobin level < 130 g/L for men and < 120 g/L for women. It has been shown that lower haemoglobin levels are as-sociated with increased mortality and morbidity among the elderly, and in patients with chronic heart failure or myocardial infarction. The red blood cell count is a routinely evaluated parameter. Reduction of RBC may lead to a decrease in the amount of oxygen transported to each cell of the human body, and thus induce a deficit of oxygen in tissues, and anaerobic metabolism, which can have negative consequences including a decrease in left ventricular ejection fraction (1).

Troponin T (TnT) is a protein forming part of the contractile apparatus of the striated muscle. The function of TnT in all types of striated muscles is the same, but cardiac Troponin T (cTnT) is different from TnT found in skeletal muscles. Therefore, cTnT detected in plasma is a highly specific marker of myocardial damage (necro-sis). High -sensitivity troponin tests, available for the past several years, detect troponin lev-els with a high degree of credibility (2). 

Was the relationship between the levels of RBC and Troponin T and the occurrence of the main endpoint assessed?

* Please add the following reference:

  1. doi: 5603/KP.2018.0076
  2. doi: 10.20452/pamw.4107

Reviewer 3 Report

In this study, Dr. Pintér and colleagues investigated the effect of anemia on heart failure (HF) with mildly reduced and preserved ejection fraction (HFmrEF/HFpEF). Overall, the study could be insightful. However, there are some major issues that have to be corrected and improved by the authors:

  • Please consult with a native English writer or a professional English editor to significantly improve the clarity and correctness of the grammar. This is a necessary step that has to be done by the authors. There are too many errors that impair the readability of this manuscript.
  • I strongly suggest to remove the HFrecEF part of this manuscript, OR the authors need to clearly explain why it is important to discuss in this manuscript. It is clear that this type of HF (HFrecEF) is out of the scope of this manuscript, as depicted in the title (only HFmrEF and EFpEF). 
  • If the authors decide to keep the HFrecEF, the title has to be amended. If they decide to remove it, please adjust the abstract and the text.
  • The clarity of the abstract has to be improved significantly. 
  • "By propensity score matching (PSM) 75 pairs of HFpEF-HFmrEF vs. HF with recovered EF (HFrecEF) patients were also selected w/o anemia to compare their outcome." The title does not contain anything about HFrecEF but out of sudden, this was written. This causes a lot of confusion for the readers. Please decide either to remove HFrecEF entirely or rewrite the entire manuscript to make it clear.
  • "By PSM, those with anemia showed poor and comparable outcome in HFpEF/HFmrEF and HFrecEF patients, respectively", what does this sentence mean? comparable to what? Please rephrase.
  • "adult population in developed countries, and the prevalence of HF is reaching ≥10% among the elderly"
  • "...which is a wide range based on the varying definition of the disease." please rephrase. A wide range of what?
  • "back-and-forth" could be changed with "bidirectional"?
  • "relationship between anemia and HF" please elaborate more. I think it is interesting to know how HF could cause anemia and vice versa. The authors have indicated briefly inside the bracket but they are not adequate. For example, how HF could cause bone marrow deficiency etc.
  • "Nevertheless, data about the incidence of this condition", which condition? Please clarify.
  • "Therefore, the aim of our study was to assess the independent predictors of all-cause mortality in patients with an LVEF ≥ 40%" I don't think this is the true aim of this study based on the title. I understand that the authors wanted to say that anemia is an independent predictor based on their results but maybe it doesn't need to be explicitly mentioned as an aim. 
  • "using our high-volume, single-center retrospective database" what does it mean by "high-volume"? Please clarify.
  • "...and to compare them to those with recovered LVEF (HFrecEF) after cardiac resynchronization therapy (CRT) implantation." as I mentioned above, why? The authors need to explain the reason of adding this data in this manuscript. 
  • "Based on our hypothesis, anemia is strongly associated with an increased risk of all-cause mortality in both cohorts, which highlights the importance of the diagnosis and management of this condition." I don't think this sentence is necessary. Consider removing.
  • In the abstract, the authors said "Between 2013 and 2019 patients with elevated NT-proBNP, EF ≥ 40% and HF symptoms were registered.", please add "HF".
  • Please add the diagnostic criteria of HF in the methods section, including the symptoms mentioned above.
  • "We also selected HFrecEF patients from our database" if the authors decide to keep this part, please explain what HFrecEF is and the diagnostic criteria used in this study. Where is it originated from? HFrEF?
  • "There were 138 patients who proved to be responders by echocardiography one year after the device implantation", Please rephrase. This sentence is difficult to understand.
  • "had identical LVEF-served as a pool for propensity score matching" identical with what?
  • Please add the ethical clearance of the study in the methods section.
  • "(69/82)" and "(50/60)" what do they mean? IQR? Please clarify
  • I don't think the p-value in Table 1 is needed. The authors never compare the anemia subpopulation in HFmrEF vs. HFpEF anyway. Consider removing.
  • Please make sure to specify IQR when using. This kind of thing is very confusing [603/1887]. We don't know what it is about. Check other publication for a reference.
  • Section 3.5 is out of scope. Consider removing.
  • Please recheck Figure 3. The authors wrongly wrote "HFrecEF" as "HFrEF" in the Figure. Please revise. 
  • "The presence of anemia was associated with a significantly higher risk in HFpEF/HFmrEF and HFrecEF patients." risk of what?
  • "Thus, anemia can be considered as an independent predictor of all-cause mortality regardless of the type of heart failure." How could the authors come to this conclusion? What was the basis? Have the authors studied HFrEF?

In general, I strongly suggest the authors to improve the clarity of their data significantly. Although the topic is relevant to discuss, there are too many flaws that potentially confuse the readers.

Please also read carefully everything (again) and maybe the authors could ask someone that is not involved in the study to read the manuscript and see if this person could understand everything. 

Round 2

Reviewer 1 Report

I would wish to congratulate the authors for answering all my concerns sufficiently. Well-done.

Author Response

Dear Colleague, 

Thank you for your answer!

Reviewer 3 Report

Thanks for the responses. However, some issues are still present and need to be corrected:

  • "The importance of anemia in heart failure (HF) patients with ejection fraction (EF)>40% by subgroup - preserved (HFpEF), mildly reduced (HFmrEF) and the newly defined recovered EF (HFrecEF)- is scarce" I am not sure if this is grammatically correct. The importance of anemia is scarce? Please revise.
  • "we described the outcome of our HFpEF-HFmrEF cohort through the presence of anemia." This is not grammatically correct. Please rephrase. 
  • "Secondly, we compared their survival with HFrecEF patientsalso having EF≥40%." maybe it should be "...patients who also have EF≥40%" or "who are also having..."? Please check again with the native English writer.
  • "Additionally, HFrecEF patients were also selected from those HFrEF patients who underwent resynchronization and as responders reached 40%EF."
  • I think the flow of the abstract is confusing and the whole abstract needs to be reorganized. How could this sentence "First, we described the outcome of our HFpEF-HFmrEF cohort through the presence of anaemia. Secondly, we compared their survival with HFrecEF patients’ also having EF≥40%." placed before this one "Using propensity score matching (PSM), 75 pairs from the HFpEF-HFmrEF and HFrecEF groups were matched by their clinical features.". It is kind of weird to see that the matching is done after the comparison. So the authors compared what to what when assessing the survival?
  • The conclusion of the abstract is missing. Please add 1-2 sentences summarizing the study.
  • Line 54: what does it mean by "the pathophysiological factors"?
  • "thus induces a deficit of oxygen in tissues, and anaerobic metabolism, which can have negative consequences including a decrease in left ventricular ejection fraction" how does this happen? How could anaerobic respiration cause an LVEF reduction? Please elaborate. 
  • "related high HF biomarkers (such as natriuretic peptides or troponin levels" The authors missed some important keywords in this process: "ischemia" and "infarction" or "cell death" are not mentioned at all. Troponin will be released in case of myocardial infarction. Cellular hypoxia itself would not be enough to cause the release if there is no cell death / apoptosis since it is actually a component of cardiomyocyte contractile machinery.
  • "HF can cause anemia due to the congestion and related absorption deficiency" which congestion? What absorption deficiency? Please clarify.
  • How could HF cause bone marrow deficiency?
  • With regards to the relation between HF and renal dysfunction, erythropoietin must be described as well. Please add the details.
  • "Consequently, the presence of anemia alone is also associated with poor outcomes in all types of HF, regardless of the left ventricular ejection fraction (LVEF) [6,7,9–11]." I think this sentence should be placed in the preceeding paragraph (one paragraph above this one).
  • "...relevance of anaemia on outcomes over 40% of LVEF in subgroups as HFpEF, HFmrEF and the newly defined HF group with recovered EF (HFrecEF) are scarce." This is not grammatically correct. Please rephrase. 
  • Also, in the previous paragraph, the authors said that the presence of anemia alone is associated with poor outcomes in all types of HF, regardless of the LVEF. If so, then the data is not scarce as claimed by the authors. Please clarify. 
  • In Bozkurt et al. [34], they defined the new classification of HF with improved LVEF as HFimpEF. I would suggest to follow this terminology, instead of making a new one called HFrecEF. 
  • "Their outcome and prognosis may be better, than HFrEF patients, although limited data are available about the comparison with those HFmrEF cohorts, in whom LVEF is similar." why only HFmrEF? what about HFpEF? So the LVEF of HFimpEF and HFpEF are not similar?
  • "[previously HFrEF patients after cardiac resynchronization therapy (CRT) implantation] who also have >40% LVEF without anemia" so the HFrecEF patients don't have anemia? Then the comparison is pointless and not appropriate. Also, it is different than what is shown in the title. Please clarify. 
  • "Those were excluded with LVEF< 40% at any time before the inclusion, or if there was an acute event at the time of inclusion or within three months were excluded" this is not grammatically correct. Please revise.
  • "patients in whom LVEF improved by over 40% after 12 months" This is misleading. This sentence means that the improvement is >40%. Please rephrase.
  • "Based on our quality insurance..." should be "assurance".
  • IQR should be written as XXX [IQR XX-XX]. Replace "/" with "-"
  • Line 307: "and exhibitied" should be "and exhibited"
  • Line 310: "inaddition to" should be "in addition to" 
  • "until 2021 there was no known effective treatment to improve the  mortality of this patient cohort" so at the moment, there are effective treatments for those types of HF? What are those treatments? Please add. 
  • "Although the proportion of HFmrEF patients was low in our patient cohort, the characteristics of these patients differed from those with HFpEF: in terms of the ischemic etiology, the proportions of men and serum NT-proBNP levels were higher" how could the lower proportion of HFmrEF affect the characteristics of those patients? Please clarify.
  • "which was significantly higher than described in high-volume RCTs" what does it mean by "high-volume"? "large-scale" maybe?
  • "This can explain the difference in medication, as because more patients in the anemia group received furosemide (72% vs. 56%) and fewer beta-blockers (81% vs. 90%) than in the non-anemia group." I feel that this is counterintuitive and needs to be discussed. Because patients with anemia tend to have worse NYHA class, they got furosemide more frequently to reduce the congestion. However, we know that adding diuretics in anemic people results in a reduction in oxygen delivery, which causes a vicious cycle. Please comment on this.
  • "In the HF population, several pathomechanisms (e.g., congestion and related malabsorption, general inflammation and frequent use of anticoagulants) are responsible for the development of anemia" the pathophysioogy of this remains unexplained. Please add the details of the pathophysiology how HF could induce anemia through those mechanisms. Also, why HF patients receive anticoagulants? 
  • "These results also highlight the importance of describing the characteristics and the outcome of different heart failure patients by site, region or country." Why did the authors say this? Did the authors suggest that the difference with data from Lund et al. was due to regional difference? What is the basis of this notion? Please clarify. 
  • "moreover it has a negative influence on outcomes of over 40% of the left ventricular ejection fraction." this is not grammatically correct. Please rephrase.
  • "such databases’ disadvantages" what are those? Please elaborate.
  • "regardless of the type of heart failure over 40% of ejection fraction (HFpEF-HFmrEF/HFrecEF)," not grammatically correct. Please rephrase. Also, why did the authors use "/"? Is it a ratio between HFpEF-HFmrEF divided by HFrecEF?
  • "early diagnostics and treatment" of what? anemia or HF? Please clarify.

In general, lots of grammatical errors remain. Not sure how the authors or MDPI did the grammar correction but it is clear that a thorough recheck by a native English writer is needed. Perhaps, someone who is familiar with the content of this manuscript could help. 

Also, the pathophysiological basis is still lacking and needs a significant improvement. 

  •  

Round 3

Reviewer 3 Report

Thanks for the responsive feedback to my comments. I have no further remarks.